# Economic evaluation of insulin glargine compared with human insulin for youth with type 1 diabetes in Tanzania and Bangladesh

Margaret L. Prust[1]*, Christina M. Lalama[2], Sylvia Kehlenbrink[3], Olatubosun Akinnola Akinola[1], Renatus Nyarubamba[4], Ajmina Hasan Flabe[5], Abigail Foulds[2], Graham D. Ogle[6], Edna Majaliwa[4,7], Bedowra Zabeen[5], Kaushik Ramaiya[4,8], Alana Garvin[1], Jing Luo[2]

1 Clinton Health Access Initiative, Inc. (CHAI), Boston, Massachusetts, United States of America, 2 Division of General Internal Medicine, Department of Medicine, University of Pittsburgh School of Medicine, Pittsburgh, Pennsylvania, United States of America, 3 Division of Endocrinology, Diabetes, and Hypertension, Brigham and Women's Hospital, Boston, Massachusetts, United States of America, 4 Tanzania Diabetes Association, Dar es Salaam, Tanzania, 5 Paediatric Diabetes Care and Research Center, Diabetic Association of Bangladesh, Dhaka, Bangladesh, 6 Life for a Child, Diabetes Australia, Sydney, Australia, 7 Muhimbili National Hospital, Dar es salaam, Tanzania, 8 Shree Hindu Mandal Hospital, Dar es salaam, Tanzania

* mprust@clintonhealthaccess.org

## Abstract

### Background

Long-acting insulin analogues are the standard of care for Type 1 diabetes (T1D) in high-income countries but remain inaccessible in many low-resource settings. Cost-effectiveness is a key consideration for their adoption. This analysis evaluated the cost-effectiveness of biosimilar insulin glargine (IGlar) versus neutral protamine Hagedorn (NPH) insulin in youth with T1D in Tanzania and Bangladesh.

### Methods

Data from the HumAn-1 trial informed a short-term economic model comparing NPH and IGlar over 12 months. The analysis, conducted from a health systems perspective, incorporated micro-costing data for insulin, injection supplies, blood glucose monitoring, and estimated hospitalization costs for complications. Effectiveness was based on a reduction in nocturnal hypoglycemia. In both countries, we calculated incremental cost-effectiveness ratios per quality-adjusted life year (QALY) gained in scenarios that compared IGlar against NPH across delivery devices (i.e., vials, cartridges, single-use pens).

### Results

IGlar was cost-effective compared to NPH when provided in cartridges in Bangladesh at a threshold of three times gross domestic product per capita. Other delivery

**Data availability statement:** Costing model inputs are available in S1 File.

**Funding:** This work is supported by a grant to the University of Pittsburgh (G-2207-05356) from The Leona M. and Harry B. Helmsley Charitable Trust. Eli Lilly provided commodity donations and unrestricted non-salary support to the Life for a Child Program of Diabetes Australia, where one of the authors (GDO) is employed. Neither Helmsley Charitable Trust nor Eli Lilly had a role in study design, data collection and analysis, decision to publish, or preparation of the manuscript.

**Competing interests:** I have read the journal's policy and the authors of this manuscript have the following competing interests: Graham D. Ogle works for the Life for a Child Program of Diabetes Australia, which receives insulin and unrestricted non-salary support from Eli Lilly. Life for a Child does advocate for the use of analog insulins in the care of children and youth with type 1 diabetes. The other authors declare that no competing interests exist. GDO's institutional affiliation with Eli Lilly does not alter our adherence to PLOS ONE policies on sharing data and materials.

devices in Bangladesh and all options in Tanzania were not cost-effective at current prices. If offered, IGlar vials would be cost-effective in Tanzania and Bangladesh if the price introduced was no more than 32% or 134% higher, respectively, than the current prices of NPH vials. Annual per patient costs and other cost-effectiveness thresholds were explored.

## Conclusion

In low-resource settings, IGlar can be a cost-effective alternative to NPH, largely due to reduced nocturnal hypoglycemia, but the delivery device for the insulin has a major impact on the costs and cost-effectiveness comparisons.

---

## Background

The number of people worldwide living with type 1 diabetes mellitus (T1DM) is projected to increase from 8.4 million in 2021 to over 13 million in 2040 [1]. Approximately one-fifth of people living with T1DM are in low-and lower-middle-income countries (LMICs) based on modeling from 201 countries using 2021 data [1], and individuals in these countries have substantially worse outcomes due to limited access to insulin, supplies, and skilled care. The same model projects that a 10-year-old diagnosed with T1DM in a high-income country could expect to live an additional 65 years, whereas a child diagnosed at the same age in a low-income country would only be expected to live 13 additional years.

Long-acting insulin analogues, which have become the de facto standard of care in high-income countries [2], offer several advantages over basal human insulins, such as fewer nighttime hypoglycemic events and once-daily dosing [3]. Treatment of micro- and macrovascular complications of diabetes, which are consequences of prolonged or recurrent hyperglycemia, accounts for the largest share of diabetes expenditure [4]. Maintaining blood glucose levels within recommended targets in order to limit the development of diabetes-related complications is a key treatment goal of diabetes therapy [5].

Despite their potential benefits, insulin analogues remain largely inaccessible in low-resource settings due to their higher cost and limited evidence of superiority over human insulin. The World Health Organization (WHO) added long-acting insulin analogues (insulin detemir, insulin degludec and insulin glargine, and their quality-assured biosimilars) to its Essential Medicines List in 2021 for the treatment of patients with type 1 or type 2 diabetes mellitus who are at high risk of experiencing hypoglycemia with human insulin [6]. However, the WHO expert committee indicated that available evidence pointed to a minimal clinical benefit of long-acting insulin analogues over human insulin and expressed concerns about the substantially greater price of long-acting insulin analogues in this context [7]. There are few studies investigating the cost-effectiveness of long-acting insulin analogues in low- and middle-income countries. Studies in 2016 in Brazil [8] and Thailand [9] found

long-acting insulin analogues were not cost-effective, but prices vary by context and over time. Other available economic evaluations come from high-income settings [10–12].

The Human versus Analogue Insulin for Youth with Type 1 Diabetes in Low-Resource Settings (HumAn-1) is a randomized trial designed to generate high-quality evidence on whether long-acting analogue insulin reduces the risk of serious hypoglycemia and/or improves glycemic time-in-range (TIR) compared with human basal insulin regimens among youth with T1DM in Tanzania and Bangladesh [13,14]. As a sub-study to the HumAn-1 trial, we conducted an economic evaluation using trial-derived differential treatment effects and intervention costing data to estimate the cost per quality-adjusted life year (QALY) gained in each arm during the trial period.

## Methods

### Model overview

This cost-utility analysis compared biosimilar insulin glargine (IGlar) with neutral protamine hagedorn insulin (NPH) in youth with T1DM using a basal-bolus regimen. The analysis was conducted from a health systems perspective. A simple model was created in Microsoft Excel (Version 2504, Microsoft Corporation, Redmond, WA, USA) to calculate the incremental cost-effectiveness ratio (ICER), measured as cost per quality-adjusted life-years (QALYs) gained, when comparing IGlar and NPH. The model included costs associated with T1DM treatment (e.g., insulin, injection supplies, commodities for monitoring blood glucose), and costs associated with acute clinical complications (e.g., hospitalizations for hypoglycemia or hyperglycemia/diabetic ketoacidosis). Effectiveness was integrated into the model using statistically significant findings from the parent clinical trial (HumAn-1). Specifically, the effectiveness measure in this model was based on nocturnal hypoglycemic events as other outcomes in the HumAn-1 trial were either not significantly different between the two study arms or were not able to be converted into QALYs. The final model includes costs associated with treatment and hypoglycemic events and effectiveness in terms of QALYs, generated by applying a disutility per nocturnal hypoglycemic event. As the time horizon for the analysis was one year, no discounting was applied. Model inputs and assumptions are described in this paper, and more detail is available in S1 File.

### Cost data

We used a micro-costing approach to estimate the annual per person cost. Direct medical costs included the sum of pharmacy costs and costs related to treatment of severe (i.e., requiring assistance from a third party) and non-severe hypoglycemic events. We assumed that all other costs were equivalent in both treatment groups.

### Cost of insulin and injection supplies

Costs for insulin and injection commodities were derived from retail prices from pharmacies in Tanzania and Bangladesh (Table 1), as procurement pricing data was not available. In Bangladesh, data was collected in eight physical and one web-based pharmacies. The manufacturer's suggested retail price (MSRP) was clearly printed on all insulins observed to be in stock and was generally the price offered, so this is what is reflected in the model. In Tanzania, data was collected from 10 pharmacies in Mwanza and 10 pharmacies in Dar Es Salaam. MSRP was not always visible, and prices varied more; the model relies on the lowest price observed in two or more pharmacies. Pharmacy visits occurred in February 2023 in Tanzania and September 2023 Bangladesh.

Pharmacies visited in both countries included a range of public hospital-based and private pharmacies in a range of neighborhoods and settings within each city. We determined the sample size by visiting additional pharmacies until we reached the point that we were not getting information on any new products or any substantially different prices. Where available, we recorded prices for all types of insulin in vials, cartridges (for reusable pens), and single-use, disposable, pens. Concentration and size of containers varied but prices were standardized for cost per 1,000 IUs. All costs reflect quality-assured products only. Costs per person per year were estimated using a body weight of 47.7 kg, the average

**Table 1. Average cost (USD) for insulin and injection supplies, by country.**

| Item | Tanzania | Bangladesh |
|---|---|---|
| *Insulin: Cost per 1000 International Units (IUs), by insulin type and delivery device* | | |
| *Vial* | | |
| Intermediate-acting human insulin (NPH) | US$ 7.43 | US$ 3.84 |
| Short-acting human insulin (Regular) | US$ 7.43 | US$ 3.84 |
| Long-acting analogue insulin (IGlar) | Not available | Not available |
| *Cartridge* | | |
| Intermediate-acting human insulin (NPH) | Not available | US$ 12.96 |
| Short-acting human insulin (Regular) | Not available | US$ 12.96 |
| Long-acting analogue insulin (IGlar) | Not available | US$ 27.61 |
| *Single-use pen* | | |
| Intermediate-acting human insulin (NPH) | US$ 37.16 | US$ 16.97 |
| Short-acting human insulin (Regular) | US$ 37.16 | US$ 16.97 |
| Long-acting analogue insulin (IGlar) | US$ 55.06 | US$ 39.49 |
| *Injection supplies: Cost per unit* | | |
| Syringe (1mL) | US$ 0.12 | US$ 0.09 |
| Alcohol swab | US$ 0.01 | US$ 0.01 |
| Reusable pen | Not available | US$ 6.48 |
| Pen needle | US$ 0.41 | US$ 0.11 |
| *Blood glucose monitoring supplies: Cost per unit* | | |
| Glucometer | US$ 20.65 | US$ 10.18 |
| Blood glucose test strip | US$ 0.33 | US$ 0.17 |
| Lancet | US$ 0.03 | US$ 0.07 |
| Alcohol swab | US$ 0.01 | US$ 0.01 |
| HbA1c test | US$ 12.39 | US$ 6.48 |

Notes: This data is based on prices observed in 2023 and the average annual exchange rate to USD for 2023. Cartridges and single-use pens are typically sold in containers of 300 IUs, but these prices have been standardized to 1000 IUs for comparability with vials.

weight of HumAn-1 trial participants at baseline. In the primary analysis, we assumed each pen needle or syringe was used four times [15], and that one alcohol swab was used for each injection.

## Cost of blood glucose monitoring

Costs for blood glucose monitoring (BGM) do not vary across the comparison groups in this model, but the costs were included because BGM represent such a large portion of annual per patient costs. The model assumed that individuals used one glucometer for the full year, performed a daily BGM testing three times daily, and received four HbA1c tests per year. One lancet and one alcohol swab was included for each instance of BGM. Costs for daily BGM testing supplies were captured using the same methods in the same pharmacies where insulin costs were recorded (**Table 1**). Costs for HbA1c testing were provided by the hospital or health center where the HumAn-1 trial took place (tests are free for youth receiving support through international insulin donation programs, but model costs reflect what adults are typically charged).

## Cost of treatment of complications and hospitalizations

Costs were estimated for five types of complications requiring differing levels of care: 1) hospitalization for diabetic ketoacidosis (DKA), 2) hospitalization for hypoglycemia complicated by the presence of an additional infection, such as malaria,

or other condition 3) hospitalization for hypoglycemia without any additional complications, 4) severe hypoglycemia without hospitalization, and 5) non-severe hypoglycemia without hospitalization (Table 2). Costs and the average number of in-patient days per type of acute complication were based on assumptions provided by the clinical and finance staff at the hospital or health center where the HumAn-1 trial took place.

*Exchange rates*: Costing data was collected in Tanzania in Bangladesh in 2023, and the average 2023 exchange rate for local currency to USD was applied (1 USD = 108.03 BDT; 1USD = 2,421.65 TZS).

## Clinical data

Clinical data were derived from the HumAn-1 trial, which compared IGlar with Standard of Care (SOC). In the SOC arm, 97.5% of participants received an NPH-based basal-bolus regimen; however, outcomes were not reported separately for those on this regimen alone. As a result, while the costing analysis reflects an NPH-based basal-bolus regimen, the clinical inputs include a small proportion of participants (2.5%) who were treated with pre-mixed human insulin (e.g., premixed 70/30).

## Insulin use

Units of insulin used per day for the IGlar and NPH groups were captured in HumAn-1 trial data. The mean (SD) total insulin dose at 12 months was 1.15 (0.44) and 1.31(0.52) units/kg/day in the IGlar and SOC arms, respectively (p value <0.001 from a multivariable generalized linear regression model adjusted for age at screening and study site) [14]. The

**Table 2. Cost (USD) and average event frequency assumptions for treatment of complications of T1DM by country.**

| Item | Average number of events per patient per year | | Cost and event frequency | |
|---|---|---|---|---|
| | SOC | IGlar | Tanzania | Bangladesh |
| *Hospitalization for DKA* | 0.025 | 0.010 | | |
| Transport * | | | US$ 82.59 | US$ 12.03 |
| In-patient days stayed per patient | | | 12 | 12 |
| Cost per in-patient day ** | | | US$ 103.24 | US$ 37.03 |
| *Hospitalization for complicated hypoglycemia* | 0.010 | 0.005 | | |
| Transport (car hire) | | | US$ 16.52 | US$ 12.03 |
| In-patient days stayed per patient | | | 6 | 6 |
| Cost per in-patient day ** | | | US$ 30.97 | US$ 37.03 |
| *Hospitalization for uncomplicated hypoglycemia* | 0.020 | 0.000 | | |
| Transport (car hire) | | | US$ 16.52 | US$ 12.03 |
| In-patient days stayed per patient | | | 1 | 1 |
| Cost per in-patient day ** | | | US$ 30.97 | US$ 37.03 |
| *Severe hypoglycemia (without hospitalization)* | 2.358 | 3.193 | | |
| Cost per additional BGM | | | US$ 0.37 | US$ 0.25 |
| Additional BGM needed per event | | | 3 | 3 |
| *Non-severe hypoglycemia (without hospitalization)* | 33.827 | 33.207 | | |
| Cost per additional BGM | | | US$ 0.37 | US$ 0.25 |
| Additional BGM needed per event | | | 2 | 2 |

**Notes:** * *Transport*: In Tanzania, transport for DKA was assumed to be by ambulance. In Bangladesh, transport for DKA was assumed to be by car hire.
** *Cost per in-patient day*: Cost per in-patient day includes typical costs for all consultations, investigations, medications, and bed fees. In Tanzania, costs reflect an ICU stay for DKA only and a general ward stay for complicated and uncompleted hypoglycemia. In Bangladesh, all hospitalizations for patients with T1DM were assumed to be treated in the ICU.

total volume of insulin use was then calculated based on weight, which was set at 47.7 kilograms, the average baseline HumAn-1 participant weight. Both regimens used a 40/60 split for basal and bolus insulin.

### Frequency of complications and hospitalizations

Table 2 shows the average number of events for each type of acute complication or hospitalization that was included in the costing as per the HumAn-1 trial. Hospitalizations are based on adverse event reporting, with adjustment for person-time in the trial. Each adverse event report was adjudicated centrally by the HumAn-1 clinical events committee, led by an independent physician to determine the category based on level of care required and the relevance of hypoglycemia as a root cause. Hypoglycemic events not requiring hospitalization were recorded based on participant self-report at each clinic or home visit and were used for the costing component only. A non-severe hypoglycemic event was defined as dizziness or confusion plus blood glucose < 3.9 mmol/L (70 mg/dl) and a severe hypoglycemic event was defined as a hypoglycemic event requiring assistance of another person to correct (but without hospitalization).

### Number of nocturnal hypoglycemia events

This cost model includes an estimate of nocturnal hypoglycemic events based on endline continuous glucose monitor (CGM) data from the HumAn-1 trial. Participants wore a CGM for the final two-week period of the HumAn-1 trial (beginning at 11.5 months after randomization), and during this time, they experienced a mean (SD) of 3.2 (2.7) and 4.2 (3.5) nocturnal hypo-glycemic events, for the IGlar and SOC arms respectively (p value = 0.001 from a multivariable linear regression model with transformed variable due to shape of distribution and adjusted for age at screening, study site, and number of events from baseline CGM) [14]. A nocturnal hypoglycemic event was defined as at least two sensor readings 15 minutes or more a part of <3.9 mmol/l or 70 mg/dl in period of 2400-600h (with no intervening values ≥3.9 mmol/l or 70 mg/dl). For the model, this was annualized to 83.20 (IGlar) and 109.20 (SOC) nocturnal hypoglycemic events by multiplying by 26 (since each CGM includes data on events covering approximately 2 weeks: 52 weeks per year/ 2 weeks = 26).

### Utility data

For this model, QALYs were calculated by applying a disutility (reduction in quality of life) based on the number of nocturnal hypoglycemic events experienced in each arm. The HumAn-1 trial measured other types of hypoglycemic events and clinical outcomes, but nocturnal hypoglycemic events was the only outcome for which disutility rates are available in the literature and where the difference in the two arms was statistically significant.

The disutility incurred was based on a large-scale time trade off (TTO) study by Evans and colleagues, which measured the disutility for non-severe nocturnal hypoglycemia events at different frequencies [16,17]. This TTO study examined symptomatic nocturnal hypoglycemic events, whereas our study includes CGM-derived events (which may or may not be symptomatic). No alternative disutility values were identified, and this issue is discussed further in the study limitations and sensitivity analyses.

Further analysis by Lauridsen and colleagues quantified the effect of diminishing marginal disutility, the idea that the negative effect of each individual hypoglycemic event on quality of life diminishes as the frequency of events increases [18]. Lauridsen and colleagues fitted a log-transformed regression curve against the data collected by Evans and colleagues resulting in the following estimation of disutility: $U_d = 0.0221x^{0.3277}$, where $U_d$ is the total disutility for nocturnal hypoglycemic events and x is the number of events experienced. The number of events observed in the two arms can therefore be used to calculate disutility of 0.0941 (IGlar) and 0.1029 (SOC).

### Cost-effectiveness threshold

In 2001, the WHO proposed that for LMICs an intervention with an ICER less than three times gross domestic product (GDP) per capita could be considered cost-effective and an ICER less than GDP per capita could be considered very cost-effective [19].

This threshold is still widely used but has also been criticized for being too high, not taking budgetary constraints into account, and not adjusting for opportunity costs [20]. No singular alternative to the WHO threshold has emerged, but we considered two other options proposed. Woods and colleagues proposed thresholds of 45% and 37% of GDP per capita for Bangladesh and Tanzania, respectively, based on empirical estimates of opportunity cost [21]. Pichon-Riviere and colleagues proposed a threshold of 12–74% of GDP per capita for LMICs like Bangladesh and Tanzania based on a conceptual model accounting for health expenditures per capita and life expectancy at birth [22]. These thresholds as they apply to the GDP per capita from each country in 2023 are shown in Table 3.

### Sensitivity analyses

Targeted, one-way sensitivity analyses were conducted to assess the impact of varying key assumptions and outcomes used in the base case analysis (Table 4).

## Results

### Treatment costs

The total costs in both countries are presented in Table 5. In Tanzania, total treatment costs were estimated to be $758 per patient per year (PPPY) for NPH and regular in vials and $1,140 for IGlar in single-use pens (and regular in vials). In the IGlar single-use pen regimen, the PPPY insulin and insulin injection supplies costs were higher by $361 (+213%) and $42 (+89%), respectively, as compared to the NPH regimen, but PPPY costs for treatment of hypoglycemic events were lower by $21 (−33%) due to lower frequency of events.

In Bangladesh, total PPPY costs for NPH-based regimens were $460 (vials), $712 (cartridges), and $790 (single-use pens). IGlar regimens cost $641 (IGlar in cartridges and regular insulin in vials), $770 (IGlar and regular insulin in cartridges), and $900 (IGlar and regular insulin in single-use pens), representing an increase of 39%, 8%, and 14%, respectively, over NPH-based regimens with the same delivery device. The PPPY cost of insulin was higher with IGlar compared to NPH, increasing by $81 to $180 depending on the delivery device. This increase was partially offset by an $10 reduction in costs due to lower frequency of hypoglycemic events and, for cartridge- and pen-based regimens, an additional $13 reduction in injection-related commodity costs due to a fewer number of daily injections.

For NPH, vial-based regimens, approximately 63% and 66% of total costs in Tanzania and Bangladesh, respectively, were attributable to blood glucose monitoring supplies. This cost component did not vary by treatment regimen.

### Incremental cost-effectiveness

The incremental cost-effectiveness ratio (ICER) per QALY gained with IGlar (single-use pens) versus NPH (vials) was US$43,508 in Tanzania (Table 6). In Bangladesh, the ICERs were US$20,580, US$6,663, and US$12,573 for IGlar versus NPH in vials, cartridges and pens, respectively. The ICER for IGlar versus NPH in cartridges was below the threshold of three times GDP per capita in Bangladesh. The ICERs per QALY gained for the other available insulin regimens were above the threshold of three times GDP per capita and above the alternative thresholds considered (Table 3).

Table 3. Cost-effectiveness threshold options relative to gross domestic product (GDP) for Tanzania and Bangladesh.

| | 3x GDP per capita | 1x GDP per capita | 74% of GDP per capita | Country-specific (see note) |
|---|---|---|---|---|
| Tanzania | 3,674 | 1,225 | 906 | 453 |
| Bangladesh | 7,653 | 2,551 | 1,888 | 1,148 |
| Threshold source | WHO; This level indicates an intervention that is cost-effective [19] | WHO; This level indicates an intervention that is very cost-effective [19] | Pinchon-Riviere et al. [22] | Woods et al.: 45% for Bangladesh and 37% for Tanzania [21] |

**Table 4. Sensitivity analyses conducted.**

| Parameter | Base case | Sensitivity analysis |
|---|---|---|
| *Adjustments to costing assumptions* | | |
| Alcohol swabs | Used for each instance of SMBG and injection | Not used at all |
| Syrine needle/reuse | Used 4 times [15] | Not re-used at all |
| Total daily insulin dose | HumAn-1 trial values by arm at 12 month follow-up:<br>• SOC: 1.31 IUs/kg/day<br>• IGlar: 1.15 IUs/kg/day | HumAn-1 trial value at baseline:<br>−1.04 IUs/kg/day |
| Daily self-monitoring of blood glucose (SMBG) | 3 times per day (as per clinical guidance) | 3 times per week (as per typical practice in some low resource settings) |
| Low-end costs | Manufacturer's suggested retail price (Bangladesh) or lowest price observed in two or more pharmacies (Tanzania) | Lowest observed price for all items |
| High-end costs | Manufacturer's suggested retail price (Bangladesh) or lowest price observed in two or more pharmacies (Tanzania) | Highest observed price for all items |
| 50% of retail prices for insulin | Manufacturer's suggested retail price (Bangladesh) or lowest price observed in two or more pharmacies (Tanzania) | 50% of MSRP or retail prices for insulin (to simulate estimated procurement pricing in the absence of country-specific data) |
| *Adjustments to clinical effectiveness assumptions* | | |
| Reduce nocturnal hypoglycemic events per year in both arms | HumAn-1 trial values (annualized) by arm at 12 month follow-up:<br>• SOC: 109.2 events per year<br>• IGlar: 83.2 events per year | Assuming that 40% of events are symptomatic [26,27]<br>• SOC: 33.3 events per year<br>• IGlar: 43.7 events per year |

**Table 5. Total treatment costs per patient per year, by country.**

| Item | NPH-based regimens | | | IGlar-based regimens | | |
|---|---|---|---|---|---|---|
| **TANZANIA** | *Vials* | *Cartridges* | *Single-use pens* | *Vials* | *Cartridges* | *Single-use pens/Vials (1)* |
| Insulin | US$ 169.53 | Not available | Not available | Not available | Not available | US$ 530.25 |
| Insulin injection commodities | US$ 47.48 | Not available | Not available | Not available | Not available | US$ 89.68 |
| Glucose monitoring supplies | US$ 477.15 | Not available | Not available | Not available | Not available | US$ 477.15 |
| Treatment of hypoglycemic events | US$ 63.78 | Not available | Not available | Not available | Not available | US$ 42.47 |
| Total | **US$ 757.94** | Not available | Not available | Not available | Not available | **US$ 1,139.55** |
| *BANGLADESH* | *Vials* | *Cartridges* | *Single-use pens* | *Cartridges/Vials (2)* | *Cartridges* | *Single-use pens* |
| Insulin | US$ 87.61 | US$ 295.56 | US$ 387.05 | US$ 267.31 | US$ 376.84 | US$ 520.16 |
| Insulin injection commodities | US$ 32.94 | US$ 76.31 | US$ 63.35 | US$ 43.92 | US$ 63.64 | US$ 50.68 |
| Glucose monitoring supplies | US$ 305.20 | US$ 305.20 | US$ 305.20 | US$ 305.20 | US$ 305.20 | US$ 305.20 |
| Treatment of hypoglycemic events | US$ 34.58 | US$ 34.58 | US$ 34.58 | US$ 24.41 | US$ 24.41 | US$ 24.41 |
| Total | **US$ 460.33** | **US$ 711.65** | **US$ 790.17** | **US$ 640.84** | **US$ 770.09** | **US$ 900.45** |

Notes: Unless otherwise noted, the regular insulin for all regimens is provided in the same delivery device as the NPH or IGlar. (1) In Tanzania, the single-use pen/vial, IGlar-based regimen includes IGlar in single-use pens and regular insulin vials (due to lack of availability of regular insulin in pens). (2) In Bangladesh, the cartridge/vial, IGlar-based regimen includes IGlar in cartridges and regular insulin vials (due to lack of availability of IGlar insulin in vials).

## Estimated cost-effectiveness if glargine were offered in vials

At the time our micro-costing work was completed, quality-assured IGlar was not available in vials in Tanzania or Bangladesh. Our analysis explored the hypothetical prices at which vials of IGlar would be considered cost-effective when compared against NPH in vials for this patient population, based on the various threshold options (Table 7). Holding other

**Table 6. Incremental cost-effectiveness ratios for various available insulin regimens, by country.**

|  | Delivery device for NPH regimen | Delivery device for IGlar regimen | Cost increment | Utility increment | ICER/ QALY |
|---|---|---|---|---|---|
| Tanzania | NPH: Vials, Regular: Vials | IGlar: Pens, Regular: Vials | $381.61 | 0.0088 | 43,508 |
| Bangladesh | NPH: Vials, Regular: Vials | IGlar: Cartridges, Regular: Vials | $180.51 | 0.0088 | 20,580 |
|  | NPH: Cartridges, Regular: Cartridges | IGlar: Cartridges, Regular: Cartridges | $58.44 | 0.0088 | 6,663 |
|  | NPH: Vials, Regular: Vials | IGlar: Pens, Regular: Pens | $440.12 | 0.0088 | 50,179 |
|  | NPH: Pens, Regular: Pens | IGlar: Pens, Regular: Pens | $110.28 | 0.0088 | 12,573 |

**Table 7. Maximum price per glargine vial (10mL) considered cost-effective, according to various thresholds.**

| Threshold | Bangladesh | Tanzania |
|---|---|---|
| WHO: 3x GDP per capita | $8.97 | $9.81 |
| WHO: 1x GDP per capita | $5.24 | $8.02 |
| Pinchon-Riviere et al.: (up to 74% of GPD per capita) [22] | $4.76 | $7.79 |
| Woods et al.: (45% and 37% of GPD per capita for Bangladesh and Tanzania, respectively) [21] | $4.22 | $7.46 |
| *For comparison*: Current price of NPH vials (10mL) | $3.84 | $7.43 |

data in the model constant, IGlar can achieve cost-effectiveness compared to NPH at all thresholds for a price higher than the currently observed retail pharmacy prices for NPH in vials. In Tanzania, where NPH vials cost US$7.43, IGlar would be cost-effective if priced up to US$9.81 using the three times GDP per capita threshold. Likewise, in Bangladesh, where NPH vials cost US$3.84, IGlar would be cost-effective if priced up to US$8.97. Under alternative thresholds, for example at 45% of GDP in Bangladesh and 37% of GDP in Tanzania, glargine vials could be cost-effective at prices of $4.22 and $7.46, respectively.

## Sensitivity analyses

The one-way sensitivity analyses conducted (Table 4) demonstrate that the results are somewhat robust but sensitive to variations in inputs for certain parameters (S1 Table). In Bangladesh, the sensitivity analyses related to total daily insulin dose and reduced nocturnal hypoglycemic events caused the ICER to exceed the cost-effectiveness threshold of three times GDP per capita (at US$ 9,499 and US$ 8,996, respectively). On the other hand, in Bangladesh, the sensitivity analysis related to estimated procurement pricing, syringe/ needle reuse and using the high-end costs cause the ICER to be lower than in the base case (at US$ 2,027, US$ 3,196 and -US$ 4,337, respectively). For Tanzania, the ICERs remained above all thresholds considered for cost-effectiveness in all sensitivity analyses.

## Discussion

Evidence is limited about the cost-effectiveness of long-acting insulin analogues in low-resource settings. However, given resource constraints, such evidence is critical for decision-making around treatment guidelines. In this simple, short-term cost-effectiveness analysis, glargine was demonstrated to be cost-effective in Bangladesh when provided in cartridges, compared to NPH in cartridges according to the most generous threshold considered (three times GDP per capita). This comparison is particularly relevant in light of recent announcements by key insulin manufacturers Novo Nordisk and Eli Lilly regarding the discontinuation of certain human insulin products (including NPH) in pens [23]. Based on other thresholds and in other delivery devices, IGlar was not cost effective in Bangladesh, and IGlar was not cost-effective in Tanzania in any scenario using our base case assumptions.

Our findings show that the cost-effectiveness of IGlar differs across countries and contexts. There are several factors that may contribute to greater cost-effectiveness in Bangladesh than Tanzania. First, most of the individual items required for treatment have a higher cost in Tanzania than in Bangladesh (Table 1), possibly due to higher production or transport costs or limited market competition. Secondly, Tanzania did not have NPH in cartridges, necessitating a comparison solely with single-use pens. These single-use pens are more costly than the cartridges and reusable pens that could be compared in the Bangladesh scenario. Had cartridges been available in Tanzania, they might have presented a more cost-effective alternative.

This model also demonstrated how insulin delivery devices may have a substantial impact on costs and cost-effectiveness. In terms of cost, vial-based treatment was less expensive than cartridges and single-use pens, which were the most expensive options, particularly in Tanzania. While vial-based insulin is less expensive, dosing may be less accurate, and the administration process is more cumbersome when compared against pens [24].

Still, IGlar in vials may be a viable option to attain the potential clinical benefits of IGlar at a lower cost. At the time of data collection, vials of quality-assured IGlar were not available in either country. Using a standard three times GDP per capita threshold, our study proposes that if NPH remains at $7.43 per vial, $9.81 should be the maximum retail price at which IGlar vials can be introduced in Tanzania to remain cost-effective when compared to NPH in vials for youth with T1D. In Bangladesh, we propose that IGlar vials should be offered at a retail price of no more than $8.97 to be cost-effective. Using alternative thresholds for cost-effectiveness (e.g., country-specific thresholds that include budget concerns and opportunity costs), our study suggests that glargine vials, when compared against NPH in vials, could be cost-effective at prices of no more than $4.22 in Bangladesh and $7.46 in Tanzania.

These results highlight the importance of considering both clinical benefits and economic factors when making treatment decisions in resource-constrained settings. The reduction in nocturnal hypoglycemia events associated with IGlar use is a significant clinical advantage as deaths from hypoglycaemia are not uncommon in low resource countries due to factors including food insecurity, limited self-monitoring of blood glucose, lack of access to glucagon, and limited after-hours access to emergency services [25,26]. At the same time, the higher cost of the medication must be weighed against this benefit. Although overall costs were lower in the NPH arm, the cost of managing complications was lower in the IGlar arm. Given the limited time horizon of this study, 12 months, it should be noted that the potential cost of long-term complications of diabetes may be substantially higher over a longer period, making IGlar increasingly more cost-effective in the long term. Longer-term modelling could be an area for further investigation.

The cost data in this study show that the cost of blood glucose monitoring is the primary cost driver across both countries and may pose a greater barrier to high-quality T1DM treatment than the type of insulin used (i.e., analogue versus human). Although blood glucose monitoring costs do not vary by treatment regimen, our model included these costs to provide a more comprehensive representation of direct costs. Blood glucose monitoring using a glucometer poses other challenges aside from cost, including pain from fingersticks and concerns about privacy, which could limit glucometer use [27]. CGMs have been shown to overcome these barriers and may lead to improved health outcomes, including in youth [28]. This calls for innovative and simpler ways to monitor blood glucose for people living with insulin dependent diabetes in low resource settings. CGMs offer a more convenient alternative but would likely increase the overall costs of blood glucose monitoring, at least initially. Strategies are needed for CGM cost-reduction and market-shaping to ensure affordability and potential inclusion in health insurance plans, thereby lowering out-of-pocket costs and improving care and outcomes.

## Limitations

Several limitations of this study should be considered. First, this analysis uses utility data from a study that is focused on assessing the impact on quality of life of symptomatic nocturnal hypoglycemic events in adults [16]. No such utility studies for hypoglycemic events have been done in children. The nocturnal hypoglycemic events in our study were identified by CGM. People who experience hypoglycemic events frequently may be less likely to experience symptoms, although

these asymptomatic nocturnal events are likely to be clinically important. Still, since asymptomatic events are less likely to directly impact one's quality of life, we conducted a sensitivity analysis that included only 40% of the CGM-derived events, based on evidence about the percentage of CGM-derived events that are symptomatic [29,30]. In this sensitivity analysis, the ICER per QALY gained for glargine in vials exceeded the cost-effectiveness threshold of three times GDP per capita in Bangladesh.

Second, the cost data in this analysis was primarily based on prices observed in retail pharmacies. However, insulin costs are currently not paid by this patient population based on retail pricing levels. Insulin for youth in Tanzania and Bangladesh are typically donated by manufacturers – Eli Lilly and Novo Nordisk, through the Life For A Child (LFAC) [31] and Changing Diabetes in Children (CDiC) [32] programs, respectively. Bulk procurement prices for insulin purchased by the government or other stakeholders were not publicly available, or in some cases relevant products are not currently being procured at all by government. Therefore, we used retail prices to represent the actual costs of goods and services as they are purchased by consumers based on the only data transparently available. Importantly, the retail pricing data also represents comparable data on all supplies from the same sources. To address these limitations, we conducted sensitivity analyses to assess the robustness of our results under the assumption that procurement prices for governmental payors are 50% of retail prices. This sensitivity analysis improved cost-effectiveness considerably. Actual mark-ups are not known may vary by product.

Third, the measure of nocturnal hypoglycemia events that drove the effectiveness in this model was based on an annualized measurement of nocturnal hypoglycemia events in a two-week period at 12 months after randomization in the HumAn-1 trial. Continuous CGM data was not available in the HumAn-1 trial, so we relied on study measures at pre-specified times. The HumAn-1 trial found that there were no significant differences in the two study arms at six months after randomization [14], and this result was corroborated by other studies in similar populations [33] that found that significant differences in key outcomes between IGlar and NPH arms only emerged between 6 and 12 months of treatment. Because of the way the HumAn-1 trial results were applied in this model, the model most closely estimates expected cost-effectiveness at the start of the second year after a switch to IGlar.

Finally, the comparisons that were possible in this analysis are somewhat limited by the delivery devices that were available in each country. Tanzania had very limited availability of IGlar and of NPH and Reg in any other delivery device except vials. In Bangladesh, IGlar was not available in vials. We conducted an exploratory analysis to understand the price at which vials of IGlar would be cost-effective.

## Conclusion

Insulin glargine may represent a cost-effective treatment option versus NPH for youth with T1DM, based on benefits related to nocturnal hypoglycemia events. However, the setting and the insulin delivery device substantially influence both costs and cost-effectiveness. Based on this work, we propose key recommendations.

Policymakers should consider the potential cost-effectiveness of IGlar, particularly in cartridge or single-use pen form, when developing treatment guidelines and making procurement decisions for youth with T1D in low resource settings. At the same time, as LMIC governments increasingly shoulder more of the costs of treating NCDs, including T1DM, for their population, they must create an enabling environment for a range of insulin products and improve procurement processes and price negotiations, with a goal of driving down prices while improving access to evidence-supported, cost-effective treatments.

Healthcare providers should be aware of the potential benefits of IGlar in reducing nocturnal hypoglycemia events but also consider the economic impact on patients and budgetary impacts on public health systems.

Industry actors should explore the business case for offering quality-assured IGlar in vials or pens at lower prices as one way to expand the market for individuals that can access analog insulins. Together, these efforts could improve the cost-effectiveness of IGlar, increase treatment options for people living with diabetes, and improve health outcomes.

 

## Supporting information

**S1 Table. Targeted sensitivity analysis results.**
(DOCX)

**S1 File. Cost-effectiveness model inputs, in detail.**
(XLSX)

## Acknowledgments

We would like to express our gratitude to the participants, health workers, and other colleagues involved in the HumAn-1 trial, upon which this study is based. We also thank Jenna Mezhrahid, Eoghan Brady and Elizabeth McCarthy for input related to the methodology.

The authors alone are responsible for the views expressed in this paper and they do not necessarily represent the views, decisions or policies of the institutions with which they are affiliated.

## Author contributions

**Conceptualization:** Margaret L. Prust, Sylvia Kehlenbrink, Abigail Foulds, Graham D. Ogle, Bedowra Zabeen, Kaushik Ramaiya, Jing Luo.

**Data curation:** Margaret L. Prust, Christina M. Lalama.

**Formal analysis:** Margaret L. Prust, Christina M. Lalama.

**Funding acquisition:** Jing Luo.

**Investigation:** Margaret L. Prust, Renatus Nyarubamba, Ajmina Hasan Flabe, Abigail Foulds, Edna Majaliwa.

**Methodology:** Margaret L. Prust.

**Project administration:** Margaret L Prust, Abigail Foulds, Edna Majaliwa, Bedowra Zabeen, Kaushik Ramaiya, Jing Luo.

**Resources:** Margaret L. Prust, Bedowra Zabeen, Kaushik Ramaiya, Jing Luo.

**Software:** Margaret L. Prust.

**Supervision:** Margaret L. Prust.

**Validation:** Margaret L. Prust, Olatubosun Akinnola Akinola, Graham D. Ogle, Jing Luo.

**Visualization:** Margaret L. Prust.

**Writing – original draft:** Margaret L. Prust, Olatubosun Akinnola Akinola.

**Writing – review & editing:** Margaret L. Prust, Christina M. Lalama, Sylvia Kehlenbrink, Olatubosun Akinnola Akinola, Renatus Nyarubamba, Ajmina Hasan Flabe, Abigail Foulds, Graham D. Ogle, Edna Majaliwa, Bedowra Zabeen, Kaushik Ramaiya, Alana Garvin, Jing Luo.

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
