## [Decision Letter · Decision Letter 0]

27 Oct 2025

Thank you for submitting your manuscript to PLOS ONE. After careful consideration, we feel that it has merit but does not fully meet PLOS ONE’s publication criteria as it currently stands. Therefore, we invite you to submit a revised version of the manuscript that addresses the points raised during the review process.

**ACADEMIC EDITOR: Major revision**

We look forward to receiving your revised manuscript.

Kind regards,

Marwan Salih Al-Nimer, MD, PhD

Academic Editor

PLOS ONE

Journal Requirements:

“This work is supported by a grant to the University of Pittsburgh (G-2207-05356) from The Leona M. and Harry B. Helmsley Charitable Trust. “

3. Thank you for stating the following in the Competing Interests:

“Graham Ogle works for the Life for a Child Program of Diabetes Australia, which receives insulin and unrestricted funding from Eli Lilly.  The other authors declare that no competing interests exist.”

We note that one or more of the authors have an affiliation to the commercial funders of this research study : Life for a Child Program of Diabetes Australia

A. Please provide an amended Funding Statement declaring this commercial affiliation, as well as a statement regarding the Role of Funders in your study. If the funding organization did not play a role in the study design, data collection and analysis, decision to publish, or preparation of the manuscript and only provided financial support in the form of authors' salaries and/or research materials, please review your statements relating to the author contributions, and ensure you have specifically and accurately indicated the role(s) that these authors had in your study. You can update author roles in the Author Contributions section of the online submission form.

B. Please also provide an updated Competing Interests Statement declaring this commercial affiliation along with any other relevant declarations relating to employment, consultancy, patents, products in development, or marketed products, etc.

Reviewers' comments:

Reviewer's Responses to Questions

**Comments to the Author**

1. Is the manuscript technically sound, and do the data support the conclusions?

Reviewer #1: Partly

Reviewer #2: Yes

2. Has the statistical analysis been performed appropriately and rigorously?

Reviewer #1: Yes

Reviewer #2: Yes

3. Have the authors made all data underlying the findings in their manuscript fully available?

Reviewer #1: Yes

Reviewer #2: Yes

4. Is the manuscript presented in an intelligible fashion and written in standard English?

Reviewer #1: Yes

Reviewer #2: Yes

Reviewer #1: This paper provides what should be valuable cost-effectiveness insulin data for LMIC settings.

Line 260-261. ‘and regular in vials and $1,140 for IGlar in single-use pens (and regular in vials).’ Not clear what the three uses of ‘in’ means in this sentence.

Line 367-368. ‘Given the limited time horizon of this study, 12 months, the cost of complications could increase significantly over a longer period,’. Since the only complication for which an advantage of glargine is being modelled is hypoglycaemia, it’s not clear why complication rates would increase over time.

It was hard to understand the use of the CGM data. As far as I can understand, all symptomatic hypos are reported as part of the trial. The nocturnal hypos captured by CGM presumably include some of these events already reported by self-report. How was this accounted for?

The CGM data were only used from the last two weeks of the trial, but according to the trial protocol in BMJ Open, CGM was done 4 times after baseline. Why were these other episodes not used?

Hypoglycaemia is the main clinical outcome used but there isn’t much detail on how the data were collected were collected or their reliability. What was the definition used for self-report hypoglycaemia? How reliable is the adverse event reporting for the cause of hospitalization? For complicated hypoglycaemia, how do you know how relevant the hypoglycaemia is? Maybe these admissions are really due to the additional infection, and hypoglycaemia plays only a minor role. Why does complicated hypoglycaemia only include infections, but not other reasons for admissions?

Reviewer #2: The study addresses an important and timely topic—evaluating the cost-effectiveness of insulin glargine compared to human insulin among type 1 diabetes in low-income countries. The study has public health relevance and potential policy implications for resource-limited settings. However, several issues related to ethical approval, research integrity, and clarity of presentation should be addressed before publication.

- Please clearly indicate ethical clearance

-Provide assurance of data confidentiality and how participate privacy was protected

-Minor grammatical and stylistic revisions are needed for fluency and academic tone.

**Do you want your identity to be public for this peer review?** For information about this choice, including consent withdrawal, please see our Privacy Policy

Reviewer #1: **Yes: ** Jonathan Shaw

Reviewer #2: No

---

## [Author Response · Author response to Decision Letter 1]

5 Dec 2025

Reviewer #1 comments

1. Line 260-261. ‘and regular in vials and $1,140 for IGlar in single-use pens (and regular in vials).’ Not clear what the three uses of ‘in’ means in this sentence.

Response: Thank you for pointing out this area that may cause confusion. These references are discussing the delivery device relevant for each cost comparison, but upon review, we determined that it may be more clear if we added the word “insulin” after regular to clarify that we are talking about insulin (not regular cartridges). The text in the paper has been amended as follows:

“In Bangladesh, total PPPY costs for NPH-based regimens were $460 (vials), $712 (cartridges), and $790 (single-use pens). IGlar regimens cost $641 (IGlar in cartridges and regular insulin in vials), $770 (IGlar and regular insulin in cartridges), and $900 (IGlar and regular insulin in single-use pens)…”

2. Line 367-368. ‘Given the limited time horizon of this study, 12 months, the cost of complications could increase significantly over a longer period,’. Since the only complication for which an advantage of glargine is being modelled is hypoglycaemia, it’s not clear why complication rates would increase over time.

Response: We appreciate you drawing out this point. The statement was meant to refer to the potential cost of other longer-term complications that are not included in this relatively short-term model. To clarify this point, the text in the paper has been amended as follows:

“Given the limited time horizon of this study, 12 months, it should be noted that the potential cost of long-term complications of diabetes may be substantially higher over a longer period…”

3. It was hard to understand the use of the CGM data. As far as I can understand, all symptomatic hypos are reported as part of the trial. The nocturnal hypos captured by CGM presumably include some of these events already reported by self-report. How was this accounted for?

Response: We have made edits to clarify that the effectiveness component of the costing model is based on the CGM data for nocturnal hypoglycemic events only, thereby avoiding the potential for double-counting hypoglycemic events. You are right to point out that the HumAn-1 trial also measured symptomatic hypoglycemia events (by participant self-report, or as a study related Adverse Event), along with other outcomes. Symptomatic hypoglycemic events by self report were not included on the effectiveness side of the cost-effectiveness equation. We have added the following statement to the Methods / Model Overview:

“Specifically, the effectiveness measure in this model was based on nocturnal hypoglycemic events as other outcomes in the HumAn-1 trial were either not significantly different between the two study arms or were not able to be converted into QALYs.”

Additionally, we amended the text in the Methods / Clinical Data as follows:

“This cost model includes an estimate of nocturnal hypoglycemic events based on endline continuous glucose monitor (CGM) data from the HumAn-1 trial. Participants wore a CGM for the final two-week period of the HumAn-1 trial (beginning at 11.5 months after randomization), and during this time, they experienced a mean (SD) of 3.2 (2.7) and 4.2 (3.5) nocturnal hypoglycemic events, for the IGlar and SOC arms respectively…”

However, self-reported hypoglycemic events were included as part of the calculation of annual costs as a result of the additional costs to re-test blood glucose in response to symptomatic events. We have therefore added the following text to the Methods / Clinical Data section:

“Hypoglycemia events not requiring hospitalization are based on participant self-report and used for the costing component only.”

4. The CGM data were only used from the last two weeks of the trial, but according to the trial protocol in BMJ Open, CGM was done 4 times after baseline. Why were these other episodes not used?

Response: Thank you for raising this important issue. The HumAn-1 trial found that there were no significant differences in key outcomes at an earlier measurement timepoint prior to 12 months, likely due to a longer than expected calibration period after switching to glargine. We have added the following text to the Limitations section to discuss this further:

“The measure of nocturnal hypoglycemia events that drove the effectiveness in this model was based on an annualized measurement of nocturnal hypoglycemia events in a two-week period at 12 months after randomization in the HumAn-1 trial. Continuous CGM data was not available in the HumAn-1 trial, so we relied on study measures at pre-specified times. The HumAn-1 trial found that there were no significant differences in the two study arms at six months after randomization [14], and this result was corroborated by other studies in similar populations [33] that found that significant differences in key outcomes between IGlar and NPH arms only emerged between 6 and 12 months of treatment. Because of the way the HumAn-1 trial results were applied in this model, the model most closely estimates expected cost-effectiveness at the start of the second year after a switch to IGlar.”

5. Hypoglycaemia is the main clinical outcome used but there isn’t much detail on how the data were collected were collected or their reliability. What was the definition used for self-report hypoglycaemia?

Response: Hypoglycemia was collected by the parent HumAn-1 trial in several ways. The number of symptomatic hypoglycemic events since their last visit was reported by each participant during every in person study visit. Hypoglycemic events that rose to the level of an trial-defined Adverse Event or a Severe Adverse Event were also collected on a rolling basis on Case Report Forms. For more details about what types of data were collected, please see our detailed Protocol and the Design/Rationale paper published in BMJ Open.

In the section on “Frequency of complications and hospitalizations”, we have added the following text to provide more detail on the definition for self-reported hypoglycemia:

“Hypoglycemic events not requiring hospitalization were recorded based on participant self-report at each clinic or home visit and were used for the costing component only. A non-severe hypoglycemic event was defined as dizziness or confusion plus blood glucose < 3.9 mmol/L (70mg/dl) and a severe hypoglycemic event was defined as a hypoglycemic event requiring assistance of another person to correct (but without hospitalization).”

Additionally, in our response to comment #3 above we have now clarified that the measurement of nocturnal hypoglycemic events was based on CGM data rather than self-report. We have added the following text to the Methods / Clinical Data section to provide more detail on the definition used by the HumAn-1 trial for a nocturnal hypoglycemic event:

“A nocturnal hypoglycemic event was defined as at least two sensor readings 15 minutes or more a part of <3.9 mmol/l or 70 mg/dl during period of 2400-600h (with no intervening values ≥ 3.9 mmol/l or 70 mg/dl).”

6. How reliable is the adverse event reporting for the cause of hospitalization?

Response: Reports of adverse events were collected by local study teams on electronic Case Report Forms. These detailed forms included required elements such as clinical history, exam findings, finger stick/lab/imaging results, admission diagnoses, hospital course (if hospitalized), and response to treatment (if any). All forms required co-signature by the local site PI’s. After electronic submission of each AE or SAE form, the overall study PI (JL) was notified and events were subsequently reviewed by the clinical events committee, led by a physician (second year endocrine fellow) that was independent of the study team. The committee classified each AE by common terminology (NCI CTCAE), type, severity and attribution/cause of each adverse event. We have added the following text to the paper to explain this measure for quality control:

“Hospitalizations are based on adverse event reporting, with adjustment for person-time in the trial. Each adverse event report was adjudicated centrally by the HumAn-1 clinical events committee, led by an independent physician.”

7. For complicated hypoglycaemia, how do you know how relevant the hypoglycaemia is? Maybe these admissions are really due to the additional infection, and hypoglycaemia plays only a minor role. Why does complicated hypoglycaemia only include infections, but not other reasons for admissions?

Response: We defined five categories of complications and hospitalizations based on consultations with local clinicians in Bangladesh and Tanzania including: a) hospitalizations for DKA, b) hospitalizations for hypoglycemia with other complications, c) hospitalizations for uncomplicated hypoglycemia, d) severe hypoglycemia (without complications), and e) non-severe hypoglycemia (without hospitalization). Clinicians in Bangladesh and Tanzania indicated that hypoglycemia in the presence of or triggered by another infection or condition typically would require more time in the hospital and higher costs to treat.

Regarding the classification process and relevance of hypoglyemia, we added the following test to the Methods section:

“Each adverse event report was adjudicated centrally by the HumAn-1 clinical events committee, led by an independent physician to determine the category based on level of care required and the relevance of hypoglycemia as a root cause.”

We’ve clarified in the Methods section that our classification is based on the level of care required (underlined text has been added):

“Costs were estimated for five types of complications requiring differing levels of care”

We further clarified that the classification of complicated hypoglycemia was based on the co-occurrence of an infection or other complicating condition:

“…hospitalization for hypoglycemia complicated by the presence of an additional infection, such as malaria, or other condition…”

Reviewer #2 comments

8. The study addresses an important and timely topic—evaluating the cost-effectiveness of insulin glargine compared to human insulin among type 1 diabetes in low-income countries. The study has public health relevance and potential policy implications for resource-limited settings. However, several issues related to ethical approval, research integrity, and clarity of presentation should be addressed before publication.

Response: Thank you for this assessment. We have responded to each of your comments below.

9. Please clearly indicate ethical clearance.

Response: This economic evaluation did not engage human subjects in research. No data was collected from or about individual human subjects for the purposes of this evaluation.

This economic evaluation leveraged analytical outputs (in aggregate form) from the HumAn-1 trial, which was a clinical randomized trial. The data used from the HumAn-1 trial is available through the cited publication (currently in pre-prints). Cost data used in the model was observed in retail outlets but collection of this data did not involve human subjects.

The HumAn-1 trial (the parent study to this economic evaluation) was approved by the Institutional Review Board at the University of Pittsburgh (STUDY21110122), the National Health Research Ethics Committee (NatHREC) at the National Institute for Medical Research in Tanzania (NIMR/HQ/R.8a/Vol.IX/4265), and the Ethical Review Committee (ERC) of Diabetic Association of Bangladesh (BADAS-ERC/EC/22/405).

We had an exchange with the journal editorial staff to provide this information upon submission of the manuscript.

10. Provide assurance of data confidentiality and how participate privacy was protected.

Response: As noted above, this economic evaluation did not engage human subjects as participants. Data used was all aggregate, secondary data from HumAn-1 or publicly available pricing data. Information about how participant privacy and other rights were protected in the HumAn-1 trial are included in the protocol, the design/rationale paper and primary results paper (currently under review) for that trial.

11. Minor grammatical and stylistic revisions are needed for fluency and academic tone.

Response: We have reviewed the paper and made a number of wording changes in response to this comment. We are open to specific recommendations around areas that require additional editing.

Journal Requirements

12. Please ensure that your manuscript meets PLOS ONE's style requirements, including those for file naming. The PLOS ONE style templates can be found at https://journals.plos.org/plosone/s/file?id=wjVg/PLOSOne_formatting_sample_main_body.pdf and https://journals.plos.org/plosone/s/file?id=ba62/PLOSOne_formatting_sample_title_authors_affiliations.pdf

Response: Thank you for the opportunity to make adjustments. We have reviewed the style requirements and made adjustments accordingly to the format and affiliations. Please let us know if there are specific areas that still require attention.

13. Thank you for stating the following financial disclosure:

“This work is supported by a grant to the University of Pittsburgh (G-2207-05356) from The Leona M. and Harry B. Helmsley Charitable Trust.” Please state what role the funders took in the study. If the funders had no role, please state: "The funders had no role in study design, data collection and analysis, decision to publish, or preparation of the manuscript."

Response: It is true that the funders had no role in the study. As per this comment and other comments from the editorial board, the financial disclosure can be updated to say:

“This work is supported by a grant to the University of Pittsburgh (G-2207-05356) from The Leona M. and Harry B. Helmsley Charitable Trust. Eli Lilly provided commodity donations and unrestricted non-salary support to the Life for a Child Program of Diabetes Australia, where one of the authors (GDO) is employed. Neither Helmsley Charitable Trust nor Eli Lilly had a role in study design, data collection and analysis, decision to publish, or preparation of the manuscript. The specific role of each author is articulated in the ‘author contributions’ section."

14. Thank you for stating the following in the Competing Interests: “Graham Ogle works for the Life for a Child Program of Diabetes Australia, which receives insulin and unrestricted funding from Eli Lilly. The other authors declare that no competing interests exist.” We note that one or more of the authors have an affiliation to the commercial funders of this research study: Life for a Child Program of Diabetes Australia. Please provide an amended Funding Statement declaring this commercial affiliation, as well as a statement regarding the Role of Funders in your study. If the funding organization did not play a role in the study design, data collection and analysis, decision to publish, or preparation of the manuscript and only provided financial support in the form of authors' salaries and/or research materials, please review your statements relating to the author contributions, and ensure you have specifically and accurately indicated the role(s) that these authors had in your study. You can update author roles in the Author Contributions section of the online submission form.

A. Please also include the following statement within your amended Funding Statement.

Response: It is true that the funders had no role in the study. As per this comment and other comments from the editorial board, the financial disclosure can be updated to say:

“This work is supported by a grant to t

---

## [Editor Report · Decision Letter 1]

14 Dec 2025

Economic evaluation of insulin glargine compared with human insulin for youth with type 1 diabetes in Tanzania and Bangladesh

PONE-D-25-50356R1

Dear Dr. Margaret L Prust,

We’re pleased to inform you that your manuscript has been judged scientifically suitable for publication and will be formally accepted for publication once it meets all outstanding technical requirements.

Kind regards,

Marwan Salih Al-Nimer, MD, PhD

Academic Editor

PLOS One
---

## [Editor Report · Acceptance letter]

PONE-D-25-50356R1

PLOS One

Dear Dr. Prust,

I'm pleased to inform you that your manuscript has been deemed suitable for publication in PLOS One. Congratulations! Your manuscript is now being handed over to our production team.

Kind regards,

on behalf of

Professor Marwan Salih Al-Nimer

Academic Editor

PLOS One